# The Role of Plant-Derived Essential Oils in Eco-Friendly Crop Protection Strategies Under Drought and Salt Stress

**DOI:** 10.3390/plants14243789

**Published:** 2025-12-12

**Authors:** Ilaria Borromeo, Cristiano Giordani, Cinzia Forni

**Affiliations:** 1Department of Biology, University of Rome Tor Vergata, Via Della Ricerca Scientifica, 00133 Rome, Italy; forni@uniroma2.it; 2Instituto de Física, Universidad de Antioquia, Calle 70 No. 52-21, Medellín 050010, Colombia; cristiano.giordani@udea.edu.co; 3Grupo Productos Naturales Marinos, Facultad de Ciencias Farmacéuticas y Alimentarias, Universidad de Antioquia, Calle 70 No. 52-21, Medellín 050010, Colombia

**Keywords:** climate change, crops protection, essential oils, horticulture, resilience, sustainability

## Abstract

Essential oils (EOs) are volatile, aromatic, and hydrophobic extracts of plant origin, known for their complex chemical composition, which often includes over 300 natural molecules with low molecular weights. They are extracted from various plant organs through physical–mechanical processes or dry distillation, and their qualitative composition and quantity change depending on the species, cultivar, and environmental growth conditions. They play a key role in plants’ response to abiotic stresses, such as drought and salinity, whose effects are intensified by climate change. Several studies showed that drought and salinity can increase or decrease EO production, depending not only on the plant species but also on the severity of the stress; in fact, in many crops an enhancement of EO synthesis was often observed under mild stress, whereas moderate or severe stress reduced production. For a few years, EOs have been considered important biostimulants and bioprotectors, capable of replacing chemical pesticides in sustainable agriculture. Consequently, seed pre-treatments (e.g., seed priming or seed coating) with EOs may represent promising tools to improve germination, stress tolerance, and crop productivity under stress conditions. Nevertheless, the high costs of extraction of EOs and the little evidence collected from field experiments still limit their use in agronomic applications. The aim of this review was to gather the most important information, published over the last ten years, concerning the effects of drought and salinity on the production of EOs and their use as biostimulants. This review critically examines the available literature, highlighting a positive perspective towards the adoption of natural approaches to reduce the environmental impact of agricultural production. Current results indicate encouraging progress in the application of EOs as biostimulants; however, further studies are needed to verify their effectiveness in real agronomic environments.

## 1. Introduction

Essential oils (EOs) are substances of plant origin, known and used for centuries by several populations, whose definition is still a matter of debate. EOs are non-polar liquid solutions with a characteristic smell, consisting of hydrophobic, volatile, and aromatic compounds, generally considered as secondary metabolites [1]. Most EOs are produced by aromatic plants belonging to the Asteraceae, Lamiaceae or Labiateae, and Apiaceae families [2]. In plants, EOs are synthesized in different organs, such as flowers, leaves, roots, rhizomes, fruits, seeds, and cortex [3], and stored in secretory tissues (e.g., glandular trichomes and resin ducts, etc.) [4,5].

Being aromatic compounds, EOs are highly soluble in volatile solvents, i.e., ethanol and ether, and insoluble in water [6]; they can be extracted through steam or dry distillation processes or by mechanical removal and subsequently separated from the aqueous phase using physical methods [7,8]. EOs’ composition varies widely according to plant species [9]. It has been estimated that the complex mixtures of some EOs may contain more than 300 different molecules with low molecular weights [10], among which various bioactive compounds, e.g., terpenes, terpenoids, polyphenols, alcohols, aldehydes, ethers, ketones, esters, amines, and amides, can be identified [11,12,13]. Their synthesis is localized in the cytoplasm and plastids of plant cells.

Due to their richness and activity, these bioactive compounds are the subject of intense research (Figure 1).

## 2. Essential Oils and Their Roles in Plant Protection

Worldwide, climate change shows an increasing trend of extreme weather phenomena, and agriculture is the most affected human activity by them [14]. Under this perspective, plants may face a wide range of environmental stress conditions (i.e., temperature extremes, flooding, drought, salinity, pollutant toxicity, spread of diseases, etc.), with deep consequences on plant growth potential and survival.

In the fight for survival, EOs are not considered essential [15]; nevertheless, they may play multiple roles during the plant life cycle and their interaction with the environment (Figure 2). EOs are involved in the response to pathogens’ attack [16] (Figure 2), as well as in plant response to abiotic stresses [4]. Among the latter, drought and salt stress are considered the most dangerous ones, deserving special attention because of their impact on plant survival and, thus, on crop productivity.

Due to the severe impact of drought and salinity on agricultural systems, this review aims to gather and critically discuss current knowledge concerning the role of EOs in modulating plant responses to these stresses, as well as the potential applications of EOs in plant protection.

Despite the advances in understanding the role played by EOs in plant adaptation to stress, significant gaps remain in understanding the effects of abiotic stresses on their biosynthesis. Further research is needed to clarify how drought and salinity influence the metabolic pathways involved in EO production and what physiological mechanisms link EO synthesis to plant stress adaptation responses. Consequently, this review also aims to explain the functional roles of EOs as components of plant adaptation mechanisms under adverse environmental conditions.

## 3. Methodology

A literature survey was undertaken based on specific keywords, e.g., essential oils, climate change, abiotic stress, salt stress, drought stress, soil salinity, and plant protection. This research was conducted on various databases (Table 1) to collect the most relevant literature. The literature survey was focused on experimental works published in the last 10 years (from 2015 to present) (Table 1).

## 4. Essential Oils and Abiotic Stress

### 4.1. Drought Stress

Water stress is considered a serious environmental problem and represents a threat to global agriculture, since it compromises plant growth and development, reducing the yield and quality of major commercial crops [21].

The arid and semi-arid areas of the planet are characterized by a desolate and sterile landscape with very poor vegetation; drought-tolerant plants, adapted for living in such habitats, reproduce for short periods during the year, when environmental conditions become favorable [22,23]. These species share common characteristics that allow them to tolerate prolonged periods of drought; these adaptations include changes in their anatomy and morphology, leading to a greater capacity to store water in their stems or leaves—thickening of the leaf cuticle, reduction or total absence of leaves to reduce transpiration rates, and the ability to survive as seeds during drought times [23,24].

Plant response depends on the genotype, stage of development, and the severity of the drought play a key role in the stress response.

When exposed to drought, non-drought-tolerant species need to activate a signaling pathway that stimulates the closure of leaf stomata to reduce transpiration and dehydration of the plant, leading to a decrease in the absorption of CO_2_ and the level of NADP^+^. These alterations cause an increase in free radicals (ROS), which negatively affect growth and photosynthesis, as well as causing the loss of cell turgor. To overcome water stress, plants adopt different strategies, among which is the accumulation of osmolytes (e.g., proline and soluble sugars) is one. To counteract ROS synthesis, the cells can enhance the activity of various antioxidant enzymes, such as catalase (CAT) and superoxide dismutase (SOD), which decrease the overproduction of ROS [22]. In response to environmental stimuli, such as abiotic stress conditions, the synthesis of EOs [22] plays an important role for plant survival, like regulating water loss during drought periods [25]. Aromatic plants are one of the most drought-sensitive species; in this group, water stress causes damage not only to plant growth but also affects the production and quality of secondary metabolites, particularly EOs (Figure 3) [22,23,24].

Several studies have demonstrated a close link between drought and EO production (Table 2); water stress can significantly change the quantity and quality of EO produced by a plant, with differences that depend not only on the species but sometimes even on the cultivars [26]. Species of the genus *Salvia* represent a good example of such differences. García-Caparrós and colleagues [27] reported a different behavior in two *Salvia* species, i.e., *S. lavandulifolia* Vahl. and *S. sclarea* L., exposed to moderate drought stress (70% evapotranspiration demand (Eto)); in the latter, a decrease in EO production was observed, unlike *S. lavandulifolia*, which was not affected by water stress [27]. Vice versa, in another species, *S. officinalis* L., an increase in EO production was observed under moderate drought stress [28].

Significant changes between the level of EO produced and the severity of stress have been reported in other species, including *Mentha piperita* L., *Ocimum basilicum* L., and *Lavandula angustifolia* Mill. and *Origanum vulgare* L., where an increase in EO has always been found under moderate or severe drought conditions [29,30,31]. As observed in many species, like *M. piperita* and *O. basilicum*, an improvement in the relative concentration of EOs has been associated with a decrease in plant biomass [27,30].

Drought also has a significant impact on the chemical composition of EOs in various aromatic plants, particularly terpenoids, which show the greatest qualitative variation in response to drought [32,33,34] (Figure 4). In plants belonging to the *Thymus* genus and in basil [35], drought is associated with variations in the concentration of thymol, carvacrol, limonene, α-pinene, β-myrcene, β-caryophyllene, linalool, 1,8-cineole, camphor, and methyl chavicol (Figure 5).

Drought exposure not only modifies the quantity of EOs produced but also their chemical composition; such variability in the response depends on the species and the severity of the stress condition applied (Figure 4).

Drought is one of the main environmental stressors able to modulate the expression of genes associated with EO metabolism [35]. The perception of water stress activates a complex network of molecular signals, involving abscisic acid (ABA), reactive oxygen species (ROS), and Ca^2+^-dependent signaling, which trigger specific transcription factors responsible for changes in gene expression [36].

Among the main transcription factors activated under drought conditions are myeloblastosis-related transcription factors (MYB), basic helix-loop-helix transcription factors (bHLH), and NAC and WRKY family genes, known to regulate enzymes involved in secondary plant metabolism [37]. Thereby, these transcription factors influence the expression of genes encoding key enzymes involved in EO biosynthesis, including terpene synthase (TPS), responsible for the synthesis of mono- and sesquiterpenes; geranyl diphosphate synthase (GPPS) and farnesyl diphosphate synthase (FPPS), important in the synthesis of isoprenoid precursors; and cytochromes P450, involved in the modification of terpenoids and phenylpropanoids [38].

In aromatic plants, moderate water stress is associated with the up-regulation of TPS genes and enzymes belonging to the MEP pathway (2-C-Methyl-D-erythritol 4-phosphate), leading to an increase in monoterpenes (e.g., linalool) [39]. On the contrary, severe drought has been related to the inhibition of gene transcription, resulting in a decrease in the accumulation of EOs and alterations in their composition [39].

The overproduction of ROS is also responsible for altering the expression of genes involved in the biosynthesis of terpenoids, while the influence of ABA on stomatal closure has indirect effects on the metabolism of isoprenoid precursors [36,40].

These pieces of evidence show how the biosynthesis of EOs is closely related to the physiological and metabolic set-up of the plant, but also how drought acts as a regulator of the transcription of genes involved in the metabolism of EOs, activating or inhibiting specific pathways. The regulation of these genes represents an important adaptive strategy that allows plants to remodel their gene profile in response to water stress conditions.

**Table 2 plants-14-03789-t002:** Bibliographic survey of scientific publications in chronological order (from 2015 to 2025) concerning the relationship between drought and EOs. CTRLs = control plants.

Plant Species	Experimental Conditions	Plant Organ Tested	Effect on the Synthesis and Composition of EO	Reference
*Eucalyptus globulus* Labill.	Greenhouse	Leaves	Increase under stress. Changes in chemical composition concerning α-pinene and eucalyptol.	[41]
*Thymus daenensis* Celak	Greenhouse. Foliar application with chitosan.	Leaves and flowers	Increase under mild stress in CTRLs. Increase under mild and severe stress in chitosan-treated plants.	[25]
*Rosmarinus officinalis* L.	Botanical garden	Leaves, flowers, and fruits	Increase under moderate stress. Changes in chemical composition concerning camphor, α-thujene, and α-pinene.	[42]
*Mentha piperita* L. *Salvia lavandulifolia* Vahl. *Salvia sclarea* L. *Thymus capitatus* L. *Thymus mastichina* L. *Lavandula latifolia* Med.	Field experiment	Plant shoots	Decrease under stress only in *L. latifolia* and *S. sclarea.* No change was observed in other species.	[27]
*Mentha spicata* L.	Field experiment	Leaves	Decrease under severe stress. Changes in chemical composition concerning carvone, limonene, and 1,8-cineole.	[43]
*Salvia nemorosa* L. *Salvia reuterana* Boiss	Greenhouse. Foliar application with melatonin.	Flowering stems	Increase under moderate stress in CTRLs. Increase under moderate stress in melatonin-treated plants. Changes in chemical composition concerning β-caryophyllene and germacrene-B in *S. nemorosa*. Changes in chemical composition concerning (E)-β-ocimene, germacrene-D, and α-gurjunene in *S. reuterana.*	[44]
*Ocimum tenuiflorum* L.	Growth chamber	Leaves	Increase under stress. Changes in chemical composition concerning eugenol and methyl eugenol.	[45]
*Lavandula angustifolia* Mill. *Lavandula stricta* Del.	Pots experiment	Plant shoots	Increase under moderate stress in *L. angustifolia*. Increase under severe stress in *L. stricta.* Changes in chemical composition concerning bornyl formate, caryophyllene oxide and linalool, and camphor in *L. angustifolia*. Changes in chemical composition concerning linalool, decanal, 1-decanol, and kessane in *L. stricta*.	[46]
*Salvia officinalis* L.	Greenhouse	Leaves	Increase under moderate stress. Changes in chemical composition concerning 1,8-cineole, α-thujone, and camphor.	[28]
*Citrus* × *latifolia* Tanaka *Citrus aurantifolia* (Christ.) Swingle	Greenhouse. Foliar application with melatonin.	Leaves	Increase under moderate and severe stress in CTRLs. Increase under moderate and severe stress in melatonin-treated plants. Changes in chemical composition concerning limonene and γ-terpinene in *C. aurantifolia*. Changes in chemical composition concerning β-pinene, sabinene, limonene, and γ-terpinene in *C. latifolia.*	[47]
*Mentha piperita* L.	Growth chamber. Bacterial inoculation with *Pseudomonas simiae* WCS417r and *Bacillus amyloliquefaciens* GB03.	Plant shoots	Increase under moderate and severe stress only in CTRLs. Changes in chemical composition concerning menthone and pulegone only in CTRLs.	[29]
*Ocimum basilicum* L. *Ocimum* × *africanum* Lour. *Ocimum americanum* L.	Greenhouse	Leaves and flowers	Decrease under severe stress in *O. basilicum* and *O. americanum*. No change was observed in *O.* x *africanum.* Drought altered the entire chemical composition of the EOs extracted from the three species.	[30]
*Ocimum basilicum* L.	Growth chamber	Plant shoots	Slight increase under moderate and severe stress. Changes in chemical composition concerning eugenol and germacrene.	[48]
*Coriandrum sativum* L.	Field experiment	Seeds	Increase under stress.	[49]
*Lavandula angustifolia* Mill.	Greenhouse	Leaves and flowers	Increase under severe stress. Changes in chemical composition concerning 1,8-cineol, camphor, and borneol.	[31]
*Thymus* × *citriodorus*	Greenhouse	Plant shoots	Decrease under stress. Change in chemical composition concerning neral, geraniol, and geranial.	[50]
*Thymus vulgaris* L.	Greenhouse. Foliar application with kaolin.	Plant shoots	Increase under moderate and severe stress in CTRLs. Increase under moderate and severe stress in kaolin-treated plants.	[51]
*Cannabis sativa* L.	Greenhouse. Foliar application with nanosilicon particles.	Inflorescences (floral bracts)	Increase under moderate stress in CTRLs. Increase under moderate stress in nanosilicon-treated plants. Change in chemical composition concerning limonene, caryophyllene, β-myrcene, β-ocimene, humulene, and cannabidiol.	[52]

### 4.2. Salt Stress

Salinity is considered one of the most destructive environmental stressors, leading to a significant reduction in crop productivity and quality worldwide [21]. This issue has become increasingly serious in arid and semi-arid lands of the planet due to the growing demand for water for irrigation [53]. Areas subjected to salt stress are increasing due to natural factors (primary salinization), such as low rainfall and high evapotranspiration, and anthropogenic factors (secondary salinization) because of the use of low-quality irrigation water [54].

Soil is considered saline when the concentration of sodium chloride, but also sulfate, magnesium, and calcium ions, is high [22]. This condition leads to an alteration of the soil, changing its texture and reducing the aeration, porosity, and water conductance [22].

Salinity negatively affects plant growth and development. In the glycophytes, the main consequences of salt stress include the inhibition of stem and root growth, the reduction in new leaf production and photosynthetic pigments, and the decrease in the absorption of water and nutrients from the soil (e.g., carbon and nitrogen) [23,24].

The accumulation of sodium at toxic levels leads to ionic and osmotic stress and subsequently oxidative stress, with an overproduction of ROS; if not effectively controlled, these conditions increase plant susceptibility to diseases and, in the most severe cases, lead to the death of the organism [23].

A strategy used by plants in response to salinity involves the accumulation of compatible osmolytes to provide greater water uptake from the soil and secondary metabolites, including EOs [22].

The relationship between EOs and salt stress is rather complex, as reported by many experimental studies (Table 3). Similarly to drought, in plants exposed to salinity, EO production is also strongly dependent on plant species, as observed in many studies; salinity increase led to a decrease in EO amounts in *R. officinalis* [55] and *M. piperita* [56].

In contrast, an enhancement of EO was reported by Meftahizadeh and colleagues [57] and Farouk and colleagues [58] in *O. basilicum* under moderate and severe salt stress conditions.

Furthermore, salt stress has a highly variable correlation with EO synthesis within the same species, with variation among the cultivars, as observed by Azimzadeh and colleagues [59] in *O. vulgare*, where only the cultivar Gracile showed an increase in EO production under mild stress. On the contrary, a decrease in EO was observed in both cultivars tested, Gracile and Vulgare, under severe stress conditions. Similar behavior was also observed in various cultivars of *O. basilicum* that showed an improvement in EO at increasing salinity, except for the Dark Opal cultivar, which revealed no enhancement under stress conditions [57].

EO production seems to be related to the threshold of salt tolerance of the specific species studied. Therefore, understanding the threshold of salinity tolerance of a crop could be useful when using EOs to manage salt stress in agriculture. The specific effect of an EO also depends on the level of salinity, which could either activate protective mechanisms under low stress or induce significant damage under moderate or severe stress [59].

Moderate or high salinity can significantly reduce EO in sensitive or moderately sensitive species (Table 3), such as *Mentha longifolia* (L.) Huds., *Coriandrum sativum*, and *O. vulgare* [59,60,61], although there are exceptions, like *Momordica charantia* L., where EO levels were improved by saline conditions [62].

These studies confirm the previous statement—the interaction between salt stress and EOs is rather complex (Figure 6), and various hypotheses still need to be clarified. However, it is possible to suggest some key points. While some EOs act as biostimulants and protectors of the plant by enhancing salt tolerance, others may be negatively affected by salinity, resulting in reduced yield and changes in chemical composition [63]. Under salt stress, a change in the chemical composition of EOs has been described. Ahl and Omer [64] reported variations in the level of borneol, carvacrol, p-cymene, (E)-2-decenal, (E)-2-dodecenal, viridiflorol, 1,8-cineole, carvone, linalool, eugenol, β-phellandrene, and myristicin in various stressed crops (Figure 7 and Figure 8).

Like drought, salinity is one of the most important environmental stressors affecting the expression of genes associated with EO biosynthesis. The accumulation of salts (e.g., NaCl) in the soil leads to osmotic and ionic stress, resulting in the alteration of various signaling pathways [21,65].

The perception of salt stress activates several signaling pathways, including Salt Overly Sensitive (SOS) and Ca^2+^-dependent pathways, as well as hormonal signaling involving ABA, ethylene, jasmonic acid, and auxins, which can trigger complex transcriptional networks in plants [21,53,65]. These networks regulate various transcription factors (some of which have already been seen for drought), including NAC, MYB, and WRKY, known to modulate genes involved in the response to ionic and osmotic stress but also to increase the synthesis of secondary metabolites engaged in reducing salt stress [66].

During EO biosynthesis, these factors influence the expression of genes encoding for the TPS and for enzymes related to MEP and mevalonate (MVA) pathways, such as 1-deoxy-D-xylulose-5-phosphate synthase (DXS), 1-deoxy-D-xylulose-5-phosphate reductoisomerase (DXR), 3-hydroxy-3-methylglutaryl-CoA reductase (HMGR), and mevalonate kinase (MVK), which are important to produce isoprenoids [67,68].

Ionic stress can also affect terpenoid synthesis, leading to cellular ionic imbalance and ROS overproduction, which reduces the NADPH/NADP^+^ ratio, key molecules for the biosynthesis of volatile compounds [69].

This evidence suggests that the transcriptional regulation of genes involved in EO metabolism under salt stress is strongly connected to ionic and osmotic balance and redox control. Like drought, salinity is a strong modulator of the expression of genes involved in EO biosynthesis, combining osmotic, ionic, and hormonal networks. It represents an important strategy for plants to counteract the toxic effects of salt, improving their resilience in high-salinity environments.

**Table 3 plants-14-03789-t003:** Bibliographic survey of scientific publications in chronological order (from 2015 to 2025) concerning the relationship between exposure to salt stress and the effects on the synthesis of EOs. CTRLs = control plants.

Plant Species	Experimental Conditions	Plant Organ Tested	Effect on the Synthesis and Composition of EO	Reference
*Salvia officinalis* L.	Greenhouse	Leaves	Salt stress did not induce the synthesis of new oils. Change in chemical composition concerning 1,8-cineol, β-thujone, camphor, borneol and viridiflorol.	[63]
*Cuminum cyminum* L.	Hydroponically cultivated in a saline solution	Seeds	Decrease under severe stress. Change in chemical composition concerning β-pinene, 1-phenyl-1,2 ethanediol, and camphor.	[70]
*Rosmarinus officinalis* L.	Greenhouse. Foliar application with salicylic acid.	Leaves	Decrease under stress in CTRLs. Decrease under severe stress in salicylic acid-treated plants. Change in chemical composition concerning cineole, camphor, borneol and verbenone in CTRLs. Change in chemical composition concerning verbenone and caryophyllene oxide in treated plants.	[71]
*Rosmarinus officinalis* L.	Field trials.	Plant shoots	Decrease under stress. Change in chemical composition concerning α-pinene, eucalyptol, camphene, borneol, D-verbenone, bornyl acetate, carcyophyllene and caryophyllene oxide.	[72]
*Dracocephalum moldavica* L.	Greenhouse. Treatment with TiO_2_ NPs, solubilized in irrigation solution.	Plant shoots	Increase synthesis under moderate and severe stress in CTRLs. Decrease under stress in treated plants. Change in chemical composition concerning 1,8-cineole, myrtenol, nerol, and β-eudesmol in CTRLs. Change in chemical composition concerning 1,8-cineole, myrtenol, germacrene and linalool in treated plants.	[73]
*Ocimum basilicum* L.	Greenhouse. Plants were treated with silicon, used as a foliar spray or soil additive.	Plant shoots	Increase synthesis under stress in CTRLs. Increase synthesis in all plants treated with silicon.	[58]
*Momordica charantia* L.	Growth chamber. Foliar application with Cs-Se NPs.	Fruits	Increase synthesis under moderate and severe stress in both CTRLs and Cs-Se NPs treated plants. Change in chemical composition concerning gentisic acid, stigmasterol, and momordin, in CTRLs and treated plants.	[62]
*Mentha piperita* L.	Greenhouse. Inoculation with *Piriformospora indica*, arbuscular mycorrhizal fungi, and co-inoculation with *P. indica* and fungi.	Leaves	Decrease in synthesis under moderate and severe stress in both CTRLs and inoculated plants. Change in chemical composition concerning menthol, menthone, and limonene.	[56]
*Anethum graveolens* L.	Greenhouse. Foliar application with GA_3_, SA, and CK.	Seeds	Decrease in synthesis under severe stress in both CTRLs and treated seeds. Change in chemical composition concerning dihydrocarvone, limonene, and dillapiole.	[74]
*Anethum graveolens* L.	Greenhouse. Biochar-based nanocomposites were added to soil.	Seeds	Increase synthesis under severe stress. Change in chemical composition concerning limonene, carvone, apiol, and dillapiole.	[75]
*Aloysia citrodora* Paláu (*Lippia citriodora* Kunth)	Greenhouse. Foliar application with Se and N-Se.	Leaves	Increase EO% under moderate and severe stress in both CTRLs and treated plants.	[76]
*Origanum vulgare* L.	Greenhouse	Plant shoots	Decrease in synthesis under severe stress in *O. vulgare* subsp. *vulgare* and *gracile*. Increase at low salt stress only in the subsp. *gracile*. Change in chemical composition concerning carvacrol, thymol, terpinene, and cymene.	[59]
*Mentha longifolia* (L.) Huds.	Greenhouse	Plant shoots	Decrease synthesis under moderate and severe stress. Change in chemical composition concerning limonene and carvone.	[60]
*Mentha spicata* L. *Origanum dictamnus* L. *Origanum onites* L.	Greenhouse	Plant shoots	No change was observed in *M. spicata*. Increase under stress in *O. onites.* Decrease under stress in *O. dictamnus.* Change in chemical composition concerning limonene, carvone, 1,8-cineole, and β-caryophyllene in *M. spicata*. Change in chemical composition concerning cymene and carvacrol in *O. dictamus*. Change in chemical composition concerning carvacrol and linalool in *O. onites*.	[34]
*Salvia abrotanoides* (Kar.) Sytsma *Salvia yangii* B.T. Drew	Field experiment	Plant shoots	Decrease under moderate or severe stress in cv. PAtKH, PAbKH, and PAbAD. Increase under moderate or severe stress in PAbSM and PAbAY. Change in chemical composition concerning 1,8-cineole, camphor, and borneol.	[77]
*Rosmarinus officinalis* L.	Greenhouse. Plants treatment involved foliar application with *Thymbra spicata* extract and inoculation with arbuscular mycorrhiza.	Plant shoots	Decrease under severe stress in CTRLs. Slight increase under moderate stress e decrease under severe stress in treated plants. Change in chemical composition concerning 1,8-cineole, camphene and geranyl acetate.	[55]
*Ocimum basilicum* L.	Pots experiment	Plant shoots	Increase under severe stress (except in the cultivar *Dark opal*).	[57]
*Thymus* × *citriodorus*	Greenhouse	Plant shoots	Decrease under stress. Change in chemical composition concerning geraniol, geranial, and neral.	[50]

## 5. Future Perspective of EOs Application in Agriculture

Agricultural research is focusing on innovative approaches to address future challenges caused by climate change, global rising population, and high food consumption. To overcome these issues, the agronomic approaches include genetic improvements of crops to increase the resistance to abiotic and biotic stresses, as well as water policies and technologies for more efficient use of irrigation water [78]. However, some of these approaches are not always economically sustainable, as they are very expensive or require a long time for crop acclimation.

Consequently, research is focusing on sustainable agroecosystems, with limited use of chemical fertilizers and pesticides, favoring natural alternatives, such as the use of biostimulants and bioprotectors, which promote plant growth and improve crop yields under adverse conditions [79]. In this scenario, EOs are gaining importance as environmentally sustainable tools, being renewable resources with low toxicity and a broad spectrum of action [80].

Biostimulants used for seed treatment represent a recent innovation, and their use has shown positive results, like increased nutrient uptake and plant stress tolerance [81], improving the genetic potential and productivity of crops [82,83]. Studies conducted by Bulgari and colleagues [84] and Parađiković and colleagues [85] have demonstrated that the application of biostimulants enhances plants’ ability to withstand adverse climatic conditions by increasing both primary and secondary metabolism, where EOs represent important components.

EOs’ activity as a biostimulant has been reported by Costa et al. [83]. In their work, they treated soybean seeds with EO from cloves; this research showed that a very low concentration of EO (1.6 mL/L) improved seed germination, root length, nodulation, and yield and reduced fungal infections of biostimulated soybean plants, thus highlighting the potential use of EOs as biostimulants.

In addition, the application of strategies against the spread of plant diseases and food preservation plays an important role in the safe feeding of the population. Infestations of mites, molds, and various species of arthropods represent the main causes of deterioration of stored foods (e.g., cereals, legumes, fruits, and vegetables), resulting in loss of quality and the release of toxins [86,87]. These infestations lead to food contamination and represent a serious risk for human and animal health [87]. Nowadays, the use of chemical pesticides damages the environment, humans, and beneficial insects, contaminating water resources, soil, and the food chain; also, the prolonged use of these substances in agricultural practice has made many pathogens and insects resistant to pesticides [88]. Consequently, a lot of research is focusing on the possible utilization of substances that are safer than traditional chemical pesticides, such as biopesticides, based on EOs [89]. Recent studies have revealed that EOs interfere with vital processes in pests through neurotoxic mechanisms and show larvicidal and ovicidal properties [89]. This evidence makes EOs promising biopesticides, increasing environmental safety due to their lower toxicity to mammals and non-target organisms [90].

The possibility of combining the use of biopesticides, based on EOs, with agronomic practices aimed at strengthening the innate defenses of plants under stress conditions represents an important achievement towards sustainable agriculture and a green economy.

In this regard, pre-farming treatments can represent valuable tools to improve plant resistance to biotic and abiotic stresses. Elicitors are biotic or abiotic agents that activate the plant defense mechanisms and induce resistance to diseases and environmental stress [91]. Under this perspective, seed treatments of different types have been foreseen and practiced locally and commercially to improve seed quality and, in the meantime, to enhance crop yield under optimal and stressed environments [92]. The application of seed coating technologies by different means improves germination phases, advances phenological events, enhances physio-morphological attributes, and yield, etc. [92].

Among the seed treatments, an innovative system is represented by seed priming, a pre-sowing treatment that consists of dipping the seeds in a priming solution, followed by air drying, which prevents the emergence of the radicle; this treatment synchronizes and improves germination, seedling establishment and nutrient absorption, inducing, in the meantime, plant cross-tolerance to biotic and abiotic stresses [53,93]. Nowadays, many priming agents have been tested in different crops and are available (i.e., polyamines, NaCl, Ca(NO_3_)_2_, GA_3_, KNO_3_, etc.).

The success of seed pre-treatments in improving plant stress tolerance is determined by the persistence of stress memory, a process where environmental stresses (e.g., drought and salinity) induce various physiological and epigenetic changes that persist and enhance the response of pre-treated seedlings to future stressors [94,95].

Among the signals involved in this type of process, various components of EOs (e.g., monoterpenes and sesquiterpenes) appear to play a key role in activating stress memory [96]. Although several studies have focused on the persistence of stress memory, many aspects remain unclear. EOs could influence signal transduction networks, acting on mitogen-activated protein kinases (MAPKs), WRKY and MYB transcription factors, and jasmonic and salicylic acids [97]. At the epigenetic level, some components of EOs lead to changes in DNA methylation, histone and chromatin modifications, promoting a faster gene response to stress [98].

Both seed treatments offer an attractive option as a tool for enhancing crop establishment and development, facing the challenges caused by climate change, and may represent cost-effective and environmentally friendly methods [93,99,100]. Even though most studies concerning EOs have been focused on their antifungal, antibacterial, and insecticidal properties, recent research aims to investigate the role of EOs in promoting germination, plant growth, and increased tolerance to environmental stressors (e.g., drought and salinity) [99,100]. Among the few experimental studies available, our recent works involving *Lippia alba* EO provide a concrete example. In these studies, the EO was used as a priming agent on seeds of *Phaseolus acutifolius* L., the germinated seeds were sown in non-saline soil, and the plants were then subjected to moderate and severe salt stress. The pre-treatment of seeds significantly increased the amount of antioxidant secondary metabolites (phenolic compounds), the concentration of proline and photosynthetic pigments, the length of the shoots and roots, the biomass, the activity of many enzymes (e.g., SOD, POD, APX, PPO, and CAT), scavenger activity, and reduced the damage caused by membrane lipid peroxidation [83,101].

These studies have shown that the application of EOs can improve the enzymatic and metabolic activities of seeds, accelerating germination and seedling development, which is particularly useful in arid or semi-arid lands. Furthermore, the use of EOs as priming agents modifies several metabolic pathways of the plant, enhancing nutrient uptake, defense against pathogens, and improving crop productivity under adverse conditions [83,101]; thus, EOs are considered as multifunctional biostimulants.

In addition, EOs are very versatile tools, since they can be used as a foliar spray or applied to the soil at the root level or used to coat seeds, making them suitable for different crops. It is worth remembering that the efficacy of EOs depends on various factors, such as the plant species from which the EOs are extracted, their chemical composition, the doses used, and plant growing conditions [102] (Table 4).

The use of EOs in agriculture may represent a good tool to reduce the impact of chemical pesticides [16], which are harmful to the environment and human health; the synthesis of EOs within plant cells and their biodegradability promote their integration into sustainable agricultural practices [99]. However, the use of EOs as biostimulants is not widespread and is hindered by several problems not yet overcome, including high costs for EO extraction, the use of expensive distillation apparatus, the need for specialized staff and suitable environments to carry out the entire extraction process [91], and differences in the effectiveness between in vitro and in vivo studies [105].

The data reported in Table 4 highlight the efficacy of various EOs in improving plant growth and resistance to adverse environmental stressors but also show some limitations that require critical analysis.

Interest in EOs is part of the planning towards ecological transition in agriculture, whose aims are the reduction in the impact of agricultural practices on the environment and, in the meantime, the provision of food supply and security under climate change [100]. In this scenario, seed pre-treatment with EOs is an innovative, natural, and eco-friendly solution that can really help crop survival.

An important aspect to consider regarding the differences in EO concentrations used in the various experiments is that the concentration of EOs used for seed pre-treatment is a fundamental detail to understand their efficacy and ensure the repeatability of the results. As demonstrated by different research groups [91,93,99], the optimal concentrations for each EO vary according to the plant species treated and the type of stress applied [99,100], but also the chemical composition of the EO may affect the observed effects.

Another key point involves experimental growth conditions. The trials were conducted in various experimental conditions, e.g., controlled growth chambers [79,93,99,100], greenhouses [101,102], and open fields [91]. It is well known that growth chambers and greenhouses do not reflect the real conditions of plant growth in the open field, leading to results difficult to generalize [106]; this variability in experimental settings can affect the reliability of results, as environmental parameters can have a direct impact on the response of plants to EOs. Therefore, we still must fill the gap between experimental evidence, obtained under controlled conditions, and field data [83,91,101,102,103,104,105,107].

Finally, the use of EOs raises issues concerning ecotoxicological risks and environmental impacts. Although they are naturally derived compounds, they can cause undesirable side effects on ecosystems if used in excessive doses because of their antibacterial and antifungal properties, which can affect soil microbiota, compromising soil health and biodiversity. The large-scale cultivation of plants used to extract EOs (e.g., rosemary and thyme) is an example of intensive land use, with potential negative consequences for environmental sustainability. Future studies should consider the balance between the agronomic benefits of EOs and ecological risks, promoting approaches that minimize negative impacts on the environment.

## 6. Conclusions

The use of EOs is becoming more important in various fields of research, such as biomedicine, biotechnology, and ecology. Currently, most research focuses on the use of EOs in medicine and microbiology, highlighting the wide range of beneficial properties typical of natural extracts. In contrast, studies concerning the involvement of EOs in crop stress resistance, particularly against abiotic stresses, remain limited; this lack of information is caused by various factors, including the complexity of performing long-term field experiments.

The data reported in this review show that EOs can play a key role in plant defense against drought and salinity, showing variations in quantity and chemical composition depending on the species or cultivar considered, the severity of the stress, but also the growing conditions and the threshold of stress tolerance of the crop.

The importance of EOs in increasing resistance to these stressors makes them interesting biostimulants or bioprotectors, thanks to their richness in natural antioxidant compounds (e.g., phenols, tannins, and saponins), but also antibacterial, antiviral, and antifungal molecules, which are beneficial for crop health.

Scientific interest in EOs as crop protection tools has increased considerably, thanks to much evidence confirming their effectiveness against various pathogens. In vitro and in vivo studies have shown that EOs extracted from *Origanum*, *Thymus*, and *Allium* possess high antimicrobial activity against phytopathogenic fungi and bacteria, inducing oxidative stress, which compromises the survival of microorganisms. Furthermore, EOs can enhance the endogenous defenses of plants—they activate key enzymes related to the antioxidant response (e.g., peroxidases) and increase the expression of genes involved in signaling pathways. These mechanisms lead to improved resistance to abiotic stress.

Despite their potential, the use of EOs still has limitations due to their volatility, high variability in chemical composition, and poor persistence in the field, which can reduce their effectiveness. To overcome these criticisms, research is developing new formulations based on nanoemulsions and nanoparticles containing EOs, which are more stable and have controlled release. Meanwhile, various studies are in progress concerning possible harmful effects on non-target organisms to ensure the safe use of these new formulations.

The utilization of EOs as biostimulants represents a sustainable agricultural strategy to improve crop tolerance to drought and salinity, preserving plant yield potential under stress conditions. Although various studies have focused on the effects of EOs during seed germination, information about the effects of these plant compounds during the advanced growth stages of horticultural crops is still limited, which means that the use of EOs is not as widespread as it should be. In the future, further steps will be necessary to make the use of EOs a feasible and economic tool in agriculture, e.g., field trials will be essential to validate and confirm the efficacy of these treatments, and scaling up of the method of extraction of EOs from the plants will be needed for a wider application in seed treatment.

A final mention regards EOs and omics sciences. The application of “omics” represents an innovative frontier in phytochemical and agronomic research. These complex mixtures of volatile compounds, traditionally used for their therapeutic properties, can be analyzed using a multi-omics approach, exploiting the knowledge provided by genomics, transcriptomics, proteomics, and metabolomics. The use of metabolomics allows for the study of different cellular metabolic profiles associated with the production of EOs, identifying biomarkers related to antimicrobial or antifungal activity. Transcriptomics and proteomics highlight modifications in gene expression and protein synthesis concerning abiotic stress signaling pathways, while genomics provides the opportunity to select specific cultivars with optimal bioactive profiles. This multi-omic approach represents a useful tool to optimize plant production and select varieties enriched in bioactive molecules beneficial for therapeutic purposes and agronomic applications, such as biostimulants and bioprotective agents for plants subjected to stress.

## Figures and Tables

**Figure 1 plants-14-03789-f001:**
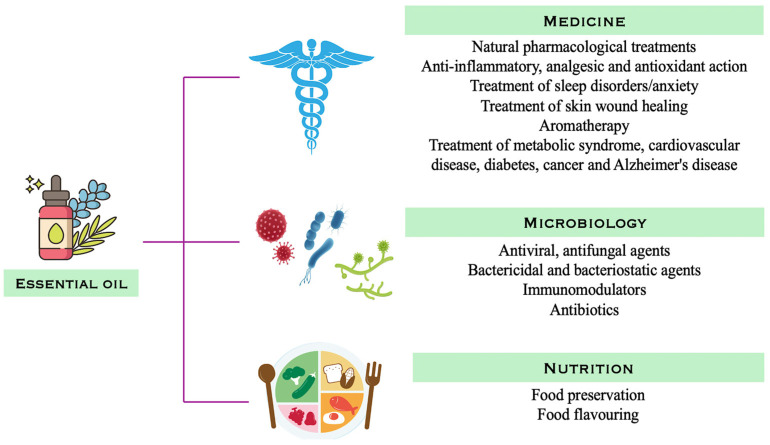
Main areas of EO use, based on their bioactivity. In the medical field, EOs are used for their anti-inflammatory, antioxidant, and analgesic properties, as well as in neurobehavioral modulation, tissue healing, and as co-adjuvants in metabolic, cardiovascular, oncological, and neurodegenerative diseases. In microbiology, they exhibit antiviral, antifungal, and antibacterial activity, with immunomodulatory effects, while in the nutritional field, they are used for food preservation and as natural flavorings.

**Figure 2 plants-14-03789-f002:**
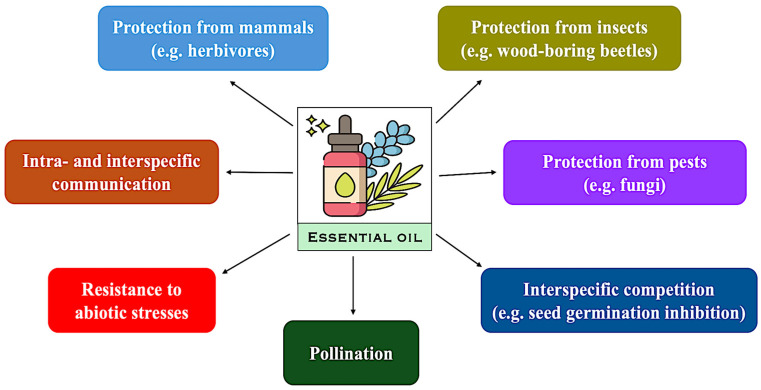
Main roles played by essential oils in plant interactions with the environment. The image was created based on information reported by [4,17,18,19,20].

**Figure 3 plants-14-03789-f003:**
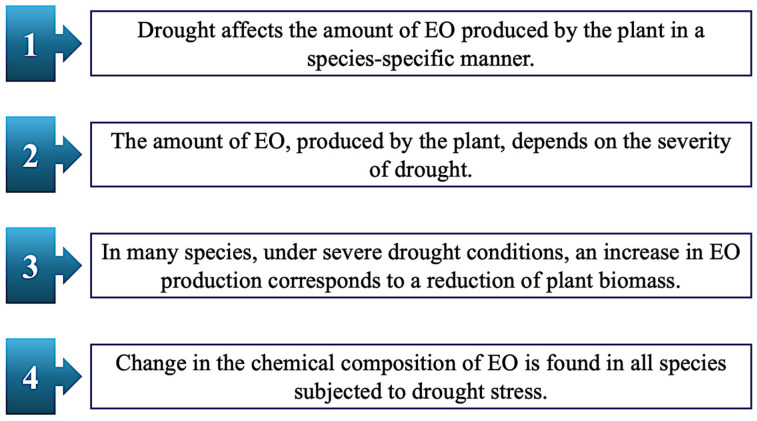
Key points concerning the effect of drought stress on the production of EO in plants.

**Figure 4 plants-14-03789-f004:**
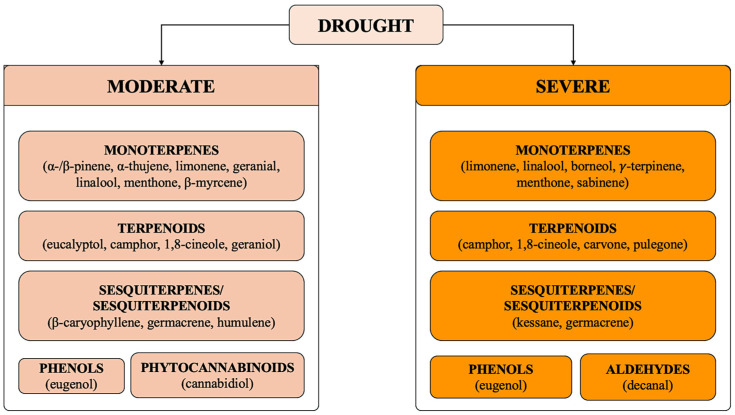
Variations in the chemical compositions of EOs induced by drought exposure. The diagram represents the major classes of chemical compounds that exhibited the greatest variations in concentration, based on the severity of drought stress.

**Figure 5 plants-14-03789-f005:**
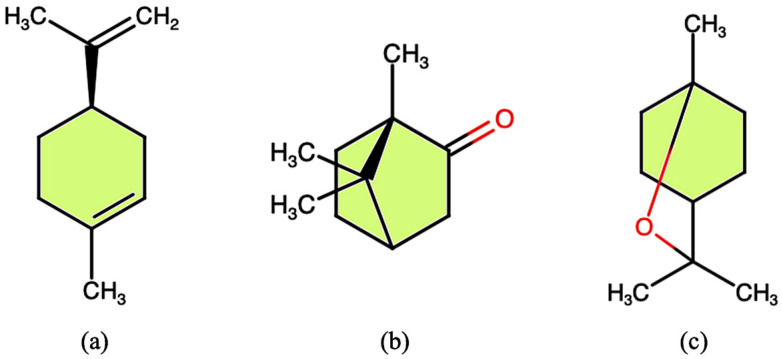
Two-dimensional chemical structure of L-limonene (**a**), DL-camphor (**b**), and 1,8-cineole (**c**), some of the molecules whose synthesis is affected by drought.

**Figure 6 plants-14-03789-f006:**
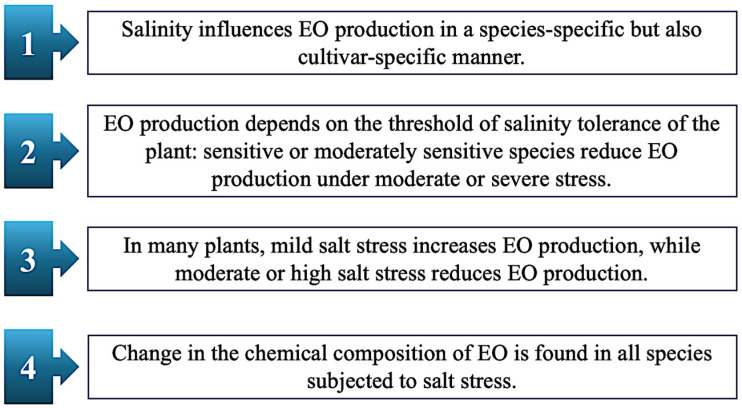
Key points concerning the effect of salt stress on the production of EO in plants.

**Figure 7 plants-14-03789-f007:**
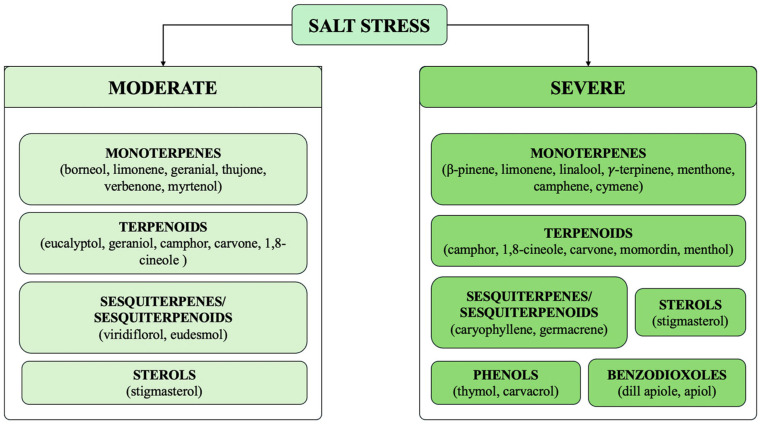
Variations in the chemical composition of EOs induced by salt stress. The diagram represents the different classes of compounds, which exhibited the greatest variations in concentration, based on the severity of salt stress.

**Figure 8 plants-14-03789-f008:**
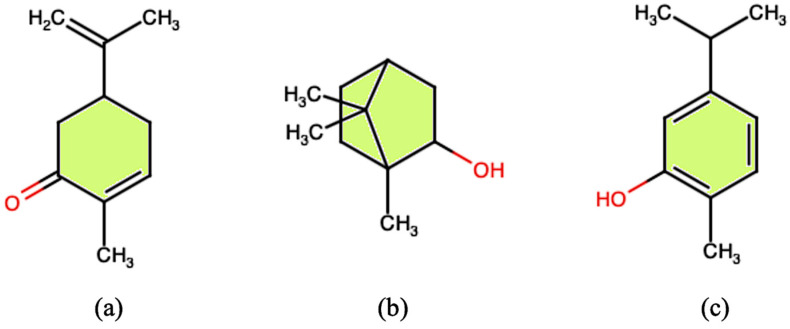
Two-dimensional chemical structure of L(-)-carvone (**a**), epi-borneol (**b**), and carvacrol (**c**), some of the molecules contained in EOs, whose synthesis is affected by salt stress.

**Table 1 plants-14-03789-t001:** Literature research strategy adopted in this review.

Item	Description
Aim of the work	To collect relevant and recent literature, exploring the relationship between EOs and plants under abiotic stress conditions.
Keywords used	essential oils, climate change, abiotic stresses, salt stress, drought stress, soil salinity, and plant protection.
Databases consulted	PubMed, Web of Science, Scopus, ScienceDirect, Google Scholar, MDPI
Type of studies included	Experimental studies
Time frame	2015–present
Inclusion criteria	Studies addressing EOs in relation to climate change and abiotic stresses (salt and drought stress)
Exclusion criteria	Studies before 2015, non-experimental works, papers not relevant to the selected keywords.

**Table 4 plants-14-03789-t004:** Bibliographic survey of scientific publications in chronological order (from 2015 to 2025) concerning the use of EOs as biostimulants. CTRLs = control plants.

Essential Oil Origin	Plant Species Exposed to Stress	Stress	Experimental Conditions and Treatment	Effect of EO Treatment on Plant	Reference
*Thymbra capitata* (L.) Cav.	*Triticum turgidum* L.	Water stress and nutrient stress	Growth chambers. Seed coating treatment.	Increase germination, shoot and roots dry weight and length, N and C content in shoots, Chl and flavonoids, of treated plants with respect to CTRLs.	[100]
*Origanum vulgare* L. *Abies alba* Mill.	*Silene sendtneri* Boiss.	No stress	Growth chambers. Seed priming treatment.	*O. vulgare* EO increased seedling length, RWC, SVI, Chl, and carotenoids of treated plants with respect to CTRLs. *A. alba* EO increased RWC, SVI, and carotenoids of treated plants with respect to CTRLs.	[93]
*Thymus capitatus* L.	*Triticum turgidum* L.	No stress	Growth chambers. Seed coating treatment.	Increase germination, shoot and root dry weight and length, amylolytic activity, and phenols of treated plants with respect to CTRLs.	[79]
*Rosmarinus officinalis* L. *Salvia officinalis* L. *Lavandula x intermedia* L.	*Triticum aestivum* L.	No stress	Growth chambers and field. Seed priming treatment.	Low EOs concentration increased germination, shoot and root length, Chl, RWC, grain yield, and grain weight of treated plants with respect to CTRLs.	[91]
*Rosmarinus officinalis* L.	*Triticum turgidum* L.	Salt stress	Environment chamber. Seed priming treatment.	High EO concentration increased germination, seedling, root, and leaf length, FW, TLA, MSI, F_v_/F_m_, carbohydrates, proline, H_2_O_2_, MDA, CAT, SOD, POD, and 7 stress-related genes of treated plants with respect to CTRLs.	[99]
*Lippia alba* Mill.	*Phaseolus acutifolius* L. *Solanum lycopersicum* L.	Salt stress	Greenhouse. Seed priming treatment.	Increase shoot and root length, biomass, phenols, flavonoids, reducing power, and scavenger activity of treated plants with respect to CTRLs, in both species.	[101]
*Syzygium aromaticum* L.	*Glycine max* L.	Salt stress during the germination stage	Germination chamber and field trial. Seed priming treatment.	Increase germination, root length, higher percentage of emergence, nodulation, and production than treated plants with industrial treatment and soybean oil controls.	[83]
*Lippia alba* Mill.	*Phaseolus acutifolius* L.	Salt stress	Greenhouse Seed priming treatment.	Increase STI, Chl, carbohydrates, proline, SOD, POD, PPO, and APX of treated plants with respect to CTRLs.	[102]
*Malva parviflora* L. (in combination with humic acid)	*Lavandula latifolia* Medik.	No stress	Field trial. Foliar spray.	Increase plant height, branch number, and plant fresh weight, and leaf area compared to CTRLs and other treatments.	[103]
*Syzygium aromaticum* L. *Origanum compactum* Bentham *Cedrus atlantica* (Endl.) Carrière *Aloysia citriodora* Palau *Salvia rosmarinus* Spenn. *Myrtus communis* L. *Thymus saturejoides* Coss. *Mentha pulegium* L.	*Cicer arietinum* L.	No stress	Germination chamber. Seed priming treatment.	Low concentration of EOs (0.01%) increases germination rate, total phenolic and flavonoid content, total soluble protein content, and mineral composition (phosphorus and sulfur content)	[104]

## Data Availability

No new data were created or analyzed in this study.

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
