# Peer review of "The Role of Plant-Derived Essential Oils in Eco-Friendly Crop Protection Strategies Under Drought and Salt Stress"

_plants, 2025, doi:10.3390/plants14243789_

Round 1
Reviewer 1 Report
Comments and Suggestions for Authors
This review contributes meaningfully to the discussion on integrating essential oils into sustainable agricultural systems and provides a valuable resource for researchers in plant physiology and agroecology. However, several significant revisions are necessary to enhance its scientific rigor, clarity, and depth of interpretation.
A statement that the search "focused on experimental studies published in the last 10 years" should indicate how review articles, meta-analyses, and conference proceedings were processed.
The introduction should more clearly formulate the specific research questions or hypotheses underlying the review. This section relies heavily on definitions and compositional descriptions, which, while informative, distract from the central issue of essential oils under abiotic stress. The authors could improve the logical flow by more clearly linking essential oil biosynthesis to stress physiology.
The caption to Figure 1 should be expanded to briefly summarize the main bioactivities of essential oils depicted in the figure.
Authors should consider summarizing key findings as a narrative synthesis, highlighting trends and inconsistencies across species rather than listing each study individually. For example, patterns associated with species-specific responses or dose-dependent essential oil production could be synthesized statistically or graphically to enhance interpretation.
Drawings of chemical structures require more precise labeling (especially for descriptions).
In the text, the authors rightly discuss the complexity of essential oil responses to stress, but their explanations remain largely descriptive. A deeper understanding of the mechanisms by which drought or salinity influences gene expression, metabolic processes, and signaling pathways would enhance the analytical depth of this review.
The section on stress memory presents an intriguing concept that deserves further development or reference to primary molecular data.
In the subsection on salt stress, some statements imply a causal relationship without sufficient evidence. These should be rephrased to reflect correlation rather than direct causation.
The consistency of units of measurement and species nomenclature in the tables should be carefully checked.
The presentation of data on changes in chemical composition would be more useful with a brief summary figure or schematic diagram linking compound classes (e.g., terpenoids, phenols) to the type and severity of stress.
A discussion of the use and effects of essential oils as biostimulants and bioprotectants requires expanded results and critical analysis.
Authors should clearly identify unresolved issues, such as scalability, formulation stability, and gaps in field validation, and provide a clearer assessment of essential oils in comparison to existing agroecological solutions.
The paragraph on seed treatments (lines 235–252) is somewhat disconnected from the previous discussion. A smoother transition and integration with the main argument is recommended.
Table 3 contains comparative data, but any discrepancies or limitations in the presented essential oil concentrations and experimental conditions should be discussed. The discussion may also include critical comments on potential environmental tradeoffs or ecotoxicological risks associated with the use of essential oils.
Strengthen conclusions by linking the results to future research directions, particularly in the context of "omics"-based approaches and field-scale validation.
Statements such as "the data contained in this work are original" are inappropriate for a review article and should be removed.
Authors should also ensure that abbreviations and terminology used in the conclusion are consistent with those previously used.
Author Response
We wish to thank the referee for the interest in our work and for the comments. The manuscript was modified according to the comments and suggestions.
This review contributes meaningfully to the discussion on integrating essential oils into sustainable agricultural systems and provides a valuable resource for researchers in plant physiology and agroecology. However, several significant revisions are necessary to enhance its scientific rigor, clarity, and depth of interpretation.
A statement that the search "focused on experimental studies published in the last 10 years" should indicate how review articles, meta-analyses, and conference proceedings were processed.
RESPONSE: The section methodology was modified, according to the comment. A table including the review research strategy and criteria adopted was added (Table 1).
The introduction should more clearly formulate the specific research questions or hypotheses underlying the review. This section relies heavily on definitions and compositional descriptions, which, while informative, distract from the central issue of essential oils under abiotic stress. The authors could improve the logical flow by more clearly linking essential oil biosynthesis to stress physiology.
RESPONSE: the section was modified and improved according to the comment (lines 78-88).
The caption to Figure 1 should be expanded to briefly summarize the main bioactivities of essential oils depicted in the figure.
RESPONSE: The caption was modified and improved according to the comment (lines 60-65).
Authors should consider summarizing key findings as a narrative synthesis, highlighting trends and inconsistencies across species rather than listing each study individually. For example, patterns associated with species-specific responses or dose-dependent essential oil production could be synthesized statistically or graphically to enhance interpretation.
RESPONSE: we are sincerely sorry, but we don’t understand the comment. What means “…synthesized statistically…”?
Drawings of chemical structures require more precise labeling (especially for descriptions).
RESPONSE: the captions were modified and improved according to the comment (lines 158-159 and 286-287).
In the text, the authors rightly discuss the complexity of essential oil responses to stress, but their explanations remain largely descriptive. A deeper understanding of the mechanisms by which drought or salinity influences gene expression, metabolic processes, and signaling pathways would enhance the analytical depth of this review.
RESPONSE: two sections were added in the text (lines 164-190 and lines 250-274).
The section on stress memory presents an intriguing concept that deserves further development or reference to primary molecular data.
RESPONSE: informations were added (lines 346-357)
In the subsection on salt stress, some statements imply a causal relationship without sufficient evidence. These should be rephrased to reflect correlation rather than direct causation.
RESPONSE: various sentences were rephrased according to the suggestion reported (from line 224 to line 243).
The consistency of units of measurement and species nomenclature in the tables should be carefully checked.
RESPONSE: checked.
The presentation of data on changes in chemical composition would be more useful with a brief summary figure or schematic diagram linking compound classes (e.g., terpenoids, phenols) to the type and severity of stress.
RESPONSE: two new figures were added (figure 4 and 7)
A discussion of the use and effects of essential oils as biostimulants and bioprotectants requires expanded results and critical analysis.
RESPONSE: informations were added (lines 300-313)
Authors should clearly identify unresolved issues, such as scalability, formulation stability, and gaps in field validation, and provide a clearer assessment of essential oils in comparison to existing agroecological solutions.
RESPONSE: the text was improved with two paragraphs (lines 378-391, 442-455).
The paragraph on seed treatments (lines 235–252) is somewhat disconnected from the previous discussion. A smoother transition and integration with the main argument is recommended.
RESPONSE: a paragraph was added to the text (lines 327-332).
Table 3 contains comparative data, but any discrepancies or limitations in the presented essential oil concentrations and experimental conditions should be discussed. The discussion may also include critical comments on potential environmental tradeoffs or ecotoxicological risks associated with the use of essential oils.
RESPONSE: informations were added (lines 403-425)
Strengthen conclusions by linking the results to future research directions, particularly in the context of "omics"-based approaches and field-scale validation.
RESPONSE: new paragraphs were added (lines 442-455, 465-477)
Statements such as "the data contained in this work are original" are inappropriate for a review article and should be removed.
RESPONSE: removed.
Authors should also ensure that abbreviations and terminology used in the conclusion are consistent with those previously used.
RESPONSE: abbreviations and terminology were checked.
Reviewer 2 Report
Comments and Suggestions for Authors
First of all, I would like to congratulate the authors for their diligent work and the effort invested in compiling research from the past decade on aromatic and medicinal plants.
In addressing my response regarding my opinion of rejecting the publication of this text, I would like to note that, with some revisions based on the following suggestions, I believe it could be resubmitted for publication:
1. Upon conducting a brief review, I feel that significant articles detailing the differences in both the quantity and composition of essential oils have been overlooked.
2. The inclusion of certain diagrams, such as (4, 6, etc.), which depict the chemical structures of various substances, appears to lack relevance in such articles because they didnot show something new.
3. The fifth chapter, which discusses future applications in agriculture, requires a more comprehensive approach and should, in my view, be connected to the preceding chapters, particularly regarding the variations caused by stress in essential oil composition. Otherwise, I find this chapter to be disconnected from the overall text.
Author Response
We would like to thank the referee for their comments and suggestions for improving the review; we sincerely apologise that the work was not interesting or did not meet expectations. However, we tried to improve it, based on the comments of all the reviewers, trying to ameliorate what had already been reported, as much as possible. We hope that, with the modifications made, this work will be more suitable for subsequent publication in such an important scientific journal.
Below are the changes requested in the comments.
First of all, I would like to congratulate the authors for their diligent work and the effort invested in compiling research from the past decade on aromatic and medicinal plants.
In addressing my response regarding my opinion of rejecting the publication of this text, I would like to note that, with some revisions based on the following suggestions, I believe it could be resubmitted for publication:
- Upon conducting a brief review, I feel that significant articles detailing the differences in both the quantity and composition of essential oils have been overlooked.
RESPONSE: We thank the referee for the comment; we agree that many articles, although important, were not used in the review; however, we decided to include only open access articles, (particularly for the tables), so that any interested reader can consult them.
Nevertheless, we have tried to improve, what has already been reported by adding other articles, regarding various aspect of EOs production, metabolism and molecular regulation, to the work (references 36-40, 65-69, 79-84, 93-97).
- The inclusion of certain diagrams, such as (4, 6, etc.), which depict the chemical structures of various substances, appears to lack relevance in such articles because they didnot show something new.
RESPONSE: We understand the referee's disagreement with the decision to include these chemical structures, but in this case, we prefer not to remove Figures 5 and 7, as the other reviewers have only requested minor improvements to the description of the chemical structures.
- The fifth chapter, which discusses future applications in agriculture, requires a more comprehensive approach and should, in my view, be connected to the preceding chapters, particularly regarding the variations caused by stress in essential oil composition. Otherwise, I find this chapter to be disconnected from the overall text.
RESPONSE: Various improvements have been made to the entire section 5.
Reviewer 3 Report
Comments and Suggestions for Authors
The article discusses the use of essential oils (EO) from plants in crop-protection approaches for combating drought and salinity stress. The review compiles the most relevant studies on EO production and its interplay with plant stress adaptation, particularly drought and salinity stress.
While plant essential oils have been extracted from different aromatic plants and find broad pharmacological applications, there has been growing interest in their use as biostimulants for crop production during stress conditions.
Some suggestions and queries for improvement are discussed:
Abstract, line 28-30: The aim of this review was to gather…………..use as biostimulants.
The theme of essential oil-based biostimulants has been increasingly studied in the present time, talking about sustainable practices in crop production. While the review compiles the published studies, what is the author’s perspective while framing the article? What is the current progress in this direction? Discuss.
Methodology:
The section on methodology should combine the rationale for selecting the topic, the research methodology followed, inclusion and exclusion criteria for the literature search. In addition, reference to the relevant databases used, keywords searched for, how the literature review was compiled and executed, and the outcome of the study. All these aspects need to be discussed in some detail. A tabular representation would provide clarity and order to the section.
Considering the present trends, are there any commercial EO-based products/biostimulants developed for agricultural applications? Nowhere has this aspect been discussed.
Line 250-252: Both seed treatments offer……………
The section needs to be elaborated and discussed in detail. No reference is provided. What are the developments in EO-based seed treatment, particularly for crop protection? Please provide key examples.
Line 296-299: The importance of EOs in increasing resistance to these stressors………………..beneficial for crop health.
In this section, it is important to discuss the key studies undertaken on plant essential oils and their application for crop protection, limitations faced, and progress achieved, citing appropriate literature.
Comments on the Quality of English Language
Moderate English revisions are required
Author Response
We kindly thank the referee for the comments. The manuscript was modified according to the suggestions reported.
The article discusses the use of essential oils (EO) from plants in crop-protection approaches for combating drought and salinity stress. The review compiles the most relevant studies on EO production and its interplay with plant stress adaptation, particularly drought and salinity stress.
While plant essential oils have been extracted from different aromatic plants and find broad pharmacological applications, there has been growing interest in their use as biostimulants for crop production during stress conditions.
Some suggestions and queries for improvement are discussed:
Abstract, line 28-30: The aim of this review was to gather…………..use as biostimulants.
The theme of essential oil-based biostimulants has been increasingly studied in the present time, talking about sustainable practices in crop production. While the review compiles the published studies, what is the author’s perspective while framing the article? What is the current progress in this direction? Discuss.
RESPONSE: The entire period was rewritten according to the comment (lines 28-35).
Methodology:
The section on methodology should combine the rationale for selecting the topic, the research methodology followed, inclusion and exclusion criteria for the literature search. In addition, reference to the relevant databases used, keywords searched for, how the literature review was compiled and executed, and the outcome of the study. All these aspects need to be discussed in some detail. A tabular representation would provide clarity and order to the section.
RESPONSE: The section methodology was modified, according to the comment. A table including the review research strategy and criteria adopted was added (Table 1).
Considering the present trends, are there any commercial EO-based products/biostimulants developed for agricultural applications? Nowhere has this aspect been discussed.
RESPONSE: we thank the referee for this interesting question.
Currently, there are no marketed biostimulants, formulated exclusively with pure EOs. Despite growing scientific interest in their potential role in mitigating abiotic stress, their intrinsic characteristics (high volatility, poor water solubility, ect.) hinder their direct use in the field. Commercial products are based on complex plant extracts, containing aromatic compounds and secondary metabolites, but not pure EOs. These extracts are better suited to current technological and agronomic requirements. On the contrary, the use of pure EOs finds practical application, almost exclusively, in seed priming and seed coating treatments, which are the only pre-sowing treatments based on the direct use of EOs, to improve germination and stress tolerance during the early stages of plant's development. Consequently, the use of EOs as biostimulants, remains mainly the subject of experimental studies, while their commercial application is still under development.
A brief paragraph, regarding these limitations in agronomic use, was already included in the paper (lines 387-394, 450-455).
Line 250-252: Both seed treatments offer……………
The section needs to be elaborated and discussed in detail. No reference is provided. What are the developments in EO-based seed treatment, particularly for crop protection? Please provide key examples.
RESPONSE: the section was improved with more information and example as requested in the comment (lines 403-415).
Line 296-299: The importance of EOs in increasing resistance to these stressors………………..beneficial for crop health.
In this section, it is important to discuss the key studies undertaken on plant essential oils and their application for crop protection, limitations faced, and progress achieved, citing appropriate literature.
RESPONSE: the section was improved with more information as requested in the comment (lines 442-455).
Round 2
Reviewer 1 Report
Comments and Suggestions for Authors
The authors have significantly improved the content of the manuscript and corrected all comments. I recommend the manuscript for acceptance.
Author Response
The authors have significantly improved the content of the manuscript and corrected all comments. I recommend the manuscript for acceptance.
We wish to thank the referee for the comments and suggested changes to improve the manuscript. We are grateful for his approval for publication.
Reviewer 3 Report
Comments and Suggestions for Authors
The manuscript may be considered in the present form.
Comments on the Quality of English LanguageModerate English revisions are required
Author Response
The manuscript may be considered in the present form.
We wish to thank the referee for the comments and suggested changes to improve the manuscript. We are grateful for his approval for publication.
Moderate English revisions are required.
The English has been improved again.